# All-Glass Single-Mode Leakage Channel Microstructured Optical Fibers with Large Mode Area and Low Bending Loss

Alexander Denisov [1,*][ID], Vladislav Dvoyrin [2], Alexey Kosolapov [1], Mikhail Likhachev [1], Vladimir Velmiskin [1], Sergey Zhuravlev [1] and Sergey Semjonov [1]

[1]   Prokhorov General Physics Institute of the Russian Academy of Sciences, Dianov Fiber Optics Research Center, 119333 Moscow, Russia
[2]   Aston Institute of Photonic Technologies, Aston University, Birmingham B4 7ET, UK
[*]   Correspondence: denisov@fo.gpi.ru

**Abstract:** The paper presents the results of theoretical and experimental studies of all-glass leakage channel microstructured optical fibers (MOFs) with a large mode area and low bending losses. These MOFs contain two layers of fluorine-doped silica glass elements with a reduced refractive index, different diameters, and different distances between them. A numerical analysis of the properties of these MOFs was performed using the finite element method. The leakage losses for the fundamental and higher-order modes were calculated in the spectral range from 0.65 μm to 1.65 μm. Simulation results show that the proposed MOF design allows for single-mode guidance in the spectral range from 0.92 μm to 1.21 μm with a bending radius of down to 0.08 m. The measured losses of the fabricated MOF with a core diameter of 22.5 μm and a bending radius of 0.1 m were less than 0.1 dB/m in the spectral range from 0.9 μm to 1.5 μm. It is demonstrated that the segments of this MOF longer than 5 m are single-mode.

**Keywords:** microstructured optical fiber; photonic crystal fiber; single-mode optical fiber; large mode area fiber; leakage channel fiber; finite element method





## 1. Introduction

Currently, there is significant interest in continuous fiber lasers of several kW [1–6], which has not weakened in recent years [7–9]. This is due to the increasingly expanding fields of practical application of lasers of this power range, including fundamental science, industrial material processing, and medicine. At the same time, such lasers are being improved, their physical and technical characteristics improved, and their cost and dimensions reduced. In addition, such lasers are part of more powerful fiber lasers, which are created by combining the radiation of several (or several dozen) separate lasers [10,11].

At the same time, research continues in the field of single-mode optical fibers with a large mode area, both active, which is the basis for high-power fiber lasers, and passive, which is necessary for delivery of laser radiation to the required distance with the required beam parameters. Various types of microstructured optical fibers (MOFs), including photonic band gap fibers [12–14], Bragg fibers [15–17], and leakage channel fibers (LCFs) [18–20], can be used for these applications in addition to traditional optical fibers with a stepped refractive index profile. However, to date, only MOFs with a photonic band gap have been successfully used in continuous fiber lasers with powers greater than 1 kW [4,9]. However, the authors of [9] during the experiments had to significantly increase the diameter of the aluminum coil on which the active fiber was wound: up to 0.7 m instead of 0.2 m. This was due to unexpectedly high bending losses, the reason for which is not yet clear and is under investigation. However, this result still does not allow for the creation of a sufficiently compact high-power fiber laser based on such a MOF.

An important feature of MOFs with leakage channels is the relative simplicity of their geometric structure, which makes it fairly easy to select their parameters to achieve a

single-mode regime with a large mode area and at the same time provide low bending losses. As a standard criterion for the single-mode regime, the condition of low leakage loss for the fundamental mode (less than 0.1 dB/m) and simultaneously high leakage loss for higher-order modes (more than 1.0 dB/m) is usually accepted [18]. At the same time, for use in high-power continuous fiber lasers, only all-glass MOFs can be used, the cladding of which is formed with elements of fluorine-doped silica glass with a reduced refractive index [21–24] instead of air holes. This is necessary because at high radiation powers, the presence of air holes in the cladding will not allow effective removal of heat from the core of the optical fiber. In addition, such MOFs have no problems when they are connected (welded) to conventional optical fibers, which is typical for MOFs with air holes in the cladding, which collapse during this process, resulting in large additional losses.

However, the variants of such all-glass MOFs studied so far [21–24] were based on a hexagonal structure and hence have very limited possibilities of varying their parameters and obtaining the required characteristics. This is explained by the use of the stack and draw technique for manufacturing the preforms for these MOFs from rods of different composition. In the theoretical work [21] for the LCF7 sample, the authors managed to achieve a bending loss level of about 0.5 dB/m for a bending radius of 0.15 m, but only when bending in one of the directions; however, when bending in the orthogonal direction, these losses were higher by an order of magnitude and were about 5 dB/m. Since in practice it is impossible to strictly control the orientation of the internal structure of MOF with respect to the bending direction, the average value of leakage losses for such MOF will be very high.

The method of drilling holes in pure silica rod with subsequent insertion of fluorine-doped silica rods into them and drawing such preform into MOF with required parameters can offer much more possibilities. This paper presents the results of theoretical and experimental studies of an all-glass MOF of original design, proposed by us and theoretically studied in previous works [25,26].

## 2. Materials and Methods

This MOF has a large pure silica glass core and a cladding containing two rings of circular elements of fluorine-doped silica glass with a reduced refractive index, different diameters, and different spacing between them (Figure 1). The key feature of this design is the presence of a ring gap between the elements of the first and second rings, which allows us to expect an increase in leakage losses for higher-order modes (HOMs) compared to the losses of fundamental mode (FM) and thereby provide a single-mode regime.

The MOF core has a diameter $D_{core}$ (dashed line in Figure 1). The first ring of the cladding contains six identical circular elements with diameter $d_1$, located at a distance $\Lambda_1$ from each other. The twelve elements of the second ring of the cladding have different diameters: $d_2 > d_1$ and $d_3 \leq d_2$, wherein the elements with diameter $d_2$ are located opposite the bridges between the elements of the first ring and at a distance $\Lambda_2$ from these elements, and the elements with diameter $d_3$ are located opposite the elements of the first ring and at a distance $\Lambda_3$ from them. This variant is conventionally called MOF-18, where 18 is the total number of elements.

Such an algorithm of MOF-18 structure provides very wide opportunities to vary its various parameters ($d_1$, $d_2$, and $d_3$, as well as $\Lambda_1$, $\Lambda_2$, $\Lambda_3$, and $\Lambda_4$) in order to optimize MOF characteristics, in particular leakage losses for fundamental mode and HOMs. Determination of the best ratio of MOF-18 parameters for a particular problem is possible using a multi-objective optimization algorithm, but even with the use of a supercomputer, this may require continuous counting for several days [27].

Therefore, we chose a relatively simple task: obtaining a single-mode regime in the spectral region around λ = 1.05 μm for MOF-18 with a core diameter of $D_{core}$ = 20 μm at a fixed ratio of element diameters ($d_2/d_1$ = 1.15, $d_3/d_1$ = 1.00) and with a bending radius down to 0.08 m. At the same time, in the case of the usual hexagonal structure with two

layers of elements (i.e., also having 18 elements), it is impossible to solve such a task because only one parameter can be varied: $d_1/\Lambda_1$.

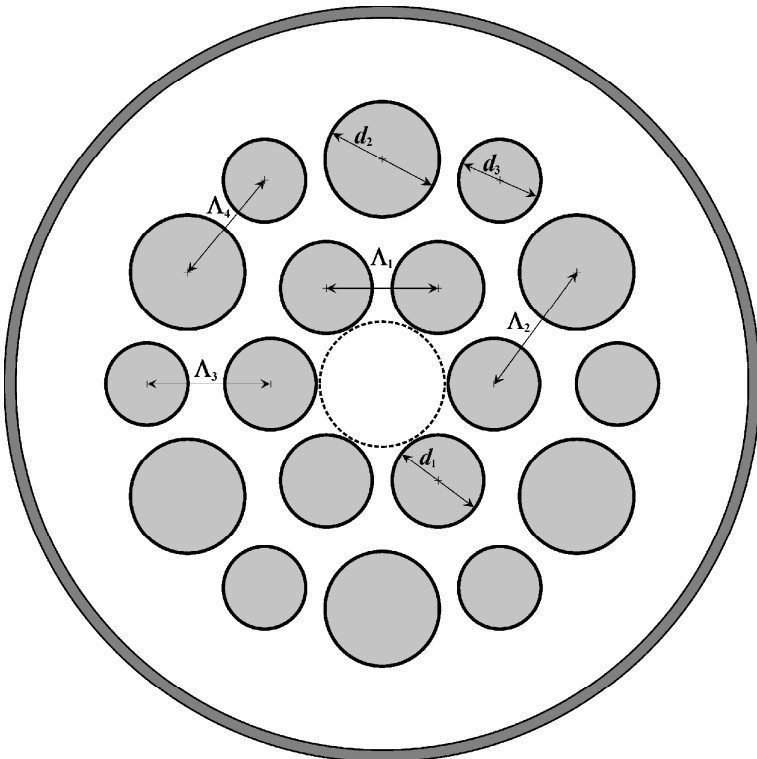

**Figure 1.** Structure of MOF-18: $d_1/\Lambda_1 = 0.82$; $d_2/d_1 = 1.25$; $d_3/d_1 = 0.90$; $\Lambda_2/\Lambda_1 = 1.25$; $\Lambda_3/\Lambda_1 = 1.10$.

The remaining MOF-18 structure parameters were determined by fitting to obtain low leakage losses for the fundamental mode (less than 0.1 dB/m) and simultaneously high leakage losses for HOMs (greater than 1.0 dB/m) over a spectral range of at least 1.0 μm to 1.1 μm, with two orthogonal bending directions considered. Thus, for calculations of the MOF-18 characteristics, we selected the following parameters: $d_1/\Lambda_1 = 0.795$, $\Lambda_2/\Lambda_1 = 1.226$, and $\Lambda_3/\Lambda_1 = 1.103$. Note that the MOF-18 variant shown in Figure 1 has slightly different parameters, chosen for the convenience of illustration.

## 3. Calculation Results

The MOF-18 characteristics were numerically calculated by the finite element method (FEM) with a cylindrical perfectly matched layer (PML) to provide the necessary accuracy of leakage losses in the case of a model structure of limited size. In Figure 1, the PML is schematically shown with a dark grey ring, and the actual PML width was determined individually for each wavelength and separately for the fundamental mode or HOMs according to the COMSOL software package manual, "Leaky Modes in a Microstructured Optical Fiber".

We employed pure silica glass as the material for MOF-18. It had refractive index $n_{sil}$, which was determined using the Sellmeier equation [28] (p. 6). For fluorine-doped silica glass elements, we set the refractive index to be $n_{fls} = n_{sil} - 4 \times 10^{-3}$.

As it was noted in [26], in MOF-18, the fundamental and higher-order modes (denoted by the number M from 1 to 6) have a group of two or three close modes for both straight and bent MOF-18, and for each group, they have the same spatial intensity distribution in the MOF-18 core. In addition, for each group, these modes have different levels of leakage loss and different real parts of their effective refractive index $n_{eff}$, so they were denoted as Ma, Mb, and Mc in descending order of their $n_{eff}$. In order to estimate the real, experimentally measured losses of any mode from the observer's point of view, additional calculations

of the integral intensities in the MOF-18 core (power flow, time average, *z* component) for all modes of these groups were performed in the present work, which were then used to determine the final losses according to a certain algorithm.

### 3.1. Straight MOF-18

Figure 2a shows the spatial intensity distributions for two polarizations of the fundamental modes (1 and 2) and four HOMs (denoted from three to six in descending order of the real part of their effective refractive index $n_{\text{eff}}$) at a wavelength of 0.850 μm for a straight MOF-18.

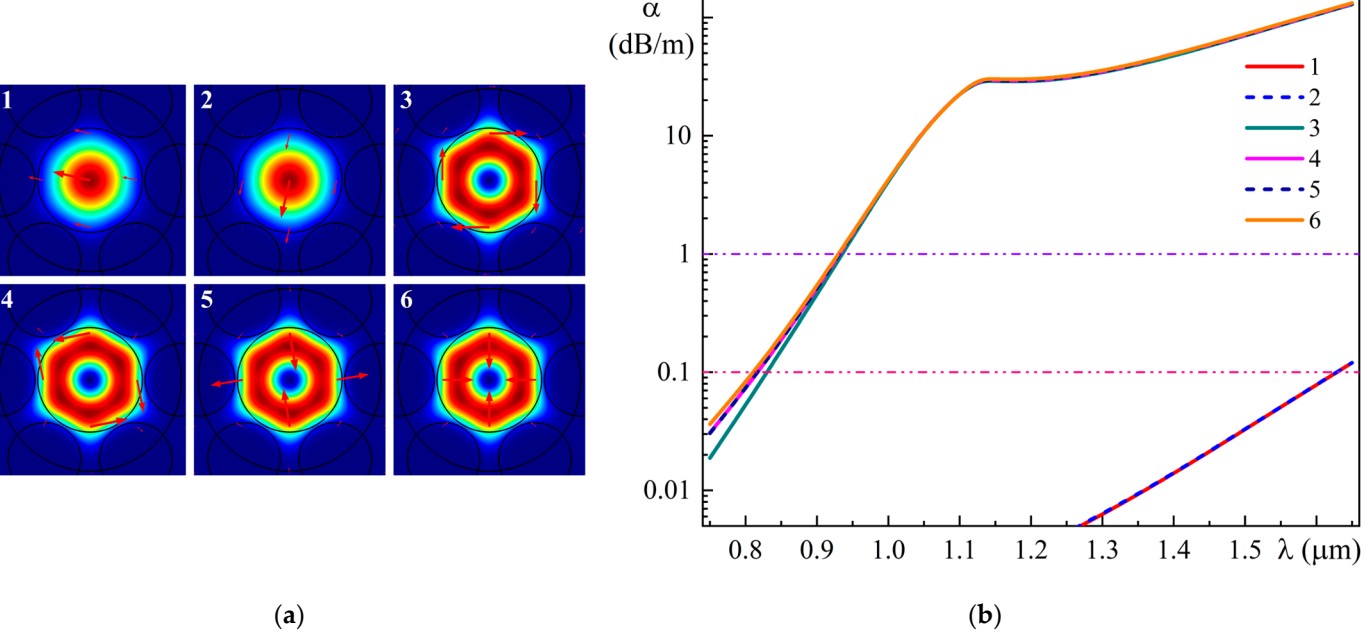

(**a**)                                                                                              (**b**)

**Figure 2.** (**a**) Spatial intensity distributions of (1, 2) fundamental modes and (3–6) higher-order modes in the straight MOF-18; (**b**) Spectral dependences of the leakage losses of (1, 2) fundamental modes and (3–6) higher-order modes in the straight MOF-18.

Modes 1 and 2 belong to the HE11 type and differ only in polarization; Mode 3 is the TE01 mode, Modes 4 and 5 are the HE21 modes, and Mode 6 is the TM01 mode [29]. Leakage losses $\alpha$ (in dB/m) were determined from the calculated imaginary part of the effective refractive index $k_{\text{eff}}$ by formula [30]

$$\alpha = \frac{20}{ln(10)} \cdot \frac{2\pi}{\lambda} k_{eff}. \tag{1}$$

Figure 2b presents the spectral dependences of leakage losses of the fundamental modes and HOMs for a straight MOF-18 in the range of 0.75–1.65 μm. In addition, this figure shows 0.1 dB/m and 1.0 dB/m loss levels to define the edges of the single-mode range. As can be seen from Figure 2b, the spectral range of the single-mode regime for the straight MOF-18 is from 0.94 μm to 1.63 μm.

Now let us consider a certain algorithm for estimating these spectral dependences, first of all, for the fundamental Modes 1 and 2. Figure 3a shows the spatial intensity distributions of Modes 1b and 2b in the straight MOF-18 for a wavelength of 1.65 μm. These modes have the same spatial intensity distributions in the MOF-18 core as Modes 1a and 2a, respectively, but relatively low values that decrease toward shorter wavelengths. In addition, Modes 1b and 2b have very high intensity values in the ring gap, while Modes 1a and 2a have very low intensity values there, so that for a standard color scheme, their intensities in the ring gap are invisible.

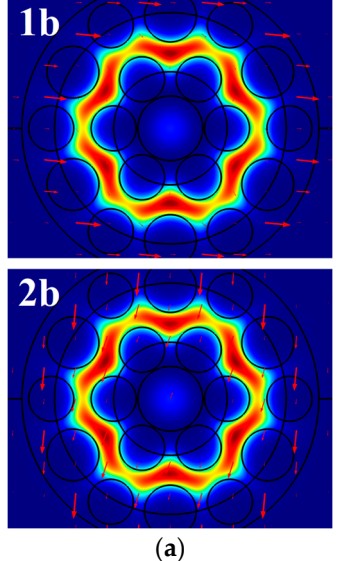

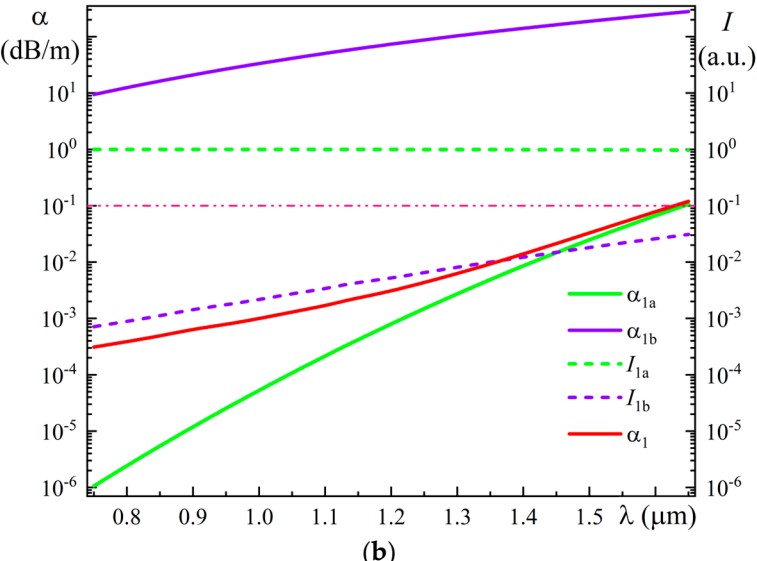

(a)

(b)

**Figure 3.** (**a**) Spatial intensity distributions of Modes 1b and 2b in the straight MOF-18 for a wavelength of 1.65 μm; (**b**) Spectral dependences of the leakage losses ($\alpha_{1a}$ and $\alpha_{1b}$) and intensities ($I_{1a}$ and $I_{1b}$) of Modes 1a and 1b in the straight MOF-18, and the resulting leakage loss for FM 1 ($\alpha_1$).

Figure 3b shows the spectral dependences of the leakage losses ($\alpha_{1a}$ and $\alpha_{1b}$) and intensities ($I_{1a}$ and $I_{1b}$) of Modes 1a and 1b in the core of straight MOF-18, with the intensities of modes normalized to their sum ($I_{1a} + I_{1b}$) for convenience.

Now suppose that a beam of radiation with wavelength λ arrives at the input of the MOF-18 segment of length *L*, and the beam has polarization corresponding to Mode 1 (coinciding with Modes 1a and 1b) and intensity $I_{01}$. It is obvious that for each particular wavelength, the incoming radiation is divided between Modes 1a and 1b in proportion to their intensities in the core, that is, $I_{01} = I_{01a} + I_{01b}$, where $I_{01a}$ and $I_{01b}$ are parts of $I_{01}$, defined by dependencies shown in Figure 3b. Then, when propagating along a section of MOF-18 of length *L*, the intensity in each of these Modes (1a and 1b) attenuates according to their respective inherent losses ($\alpha_{1a}$ and $\alpha_{1b}$).

Using the standard expression for loss in dB/m,

$$\alpha = -(10/L)\cdot\log(I/I_0), \tag{2}$$

we can write

$$I_{1a}(L) = I_{01a}\cdot10^{-\alpha(1a)\cdot L/10}, \tag{3}$$

$$I_{1b}(L) = I_{01b}\cdot10^{-\alpha(1b)\cdot L/10}, \tag{4}$$

where $\alpha(1a) \equiv \alpha_{1a}$ and $\alpha(1b) \equiv \alpha_{1b}$.

As a result, the total intensity of Mode 1 (as a sum of Modes 1a and 1b) at the output of MOF-18 segment *L* can be determined by the expression

$$I_1(L) = I_{1a}(L) + I_{1b}(L) = I_{01a}\cdot10^{-\alpha(1a)\cdot L/10} + I_{01b}\cdot10^{-\alpha(1b)\cdot L/10}. \tag{5}$$

Then, conventionally, we can determine the loss for Mode 1 using the following expression:

$$\alpha_1 = -(10/L)\cdot\log((I_{01a}\cdot10^{-\alpha(1a)\cdot L/10} + I_{01b}\cdot10^{-\alpha(1b)\cdot L/10})/(I_{01a} + I_{01b})). \tag{6}$$

It should be emphasized that this expression for the loss of Mode 1 (in dB/m) depends on the length *L* of MOF-18, which should be taken into account when using this value in practice. Since one of the main applications of MOF-18 is its use for the transmission of high-

power laser radiation, which requires segments of tens of meters in length, calculations of the spectral dependence of leakage loss $\alpha 1$ of fundamental Mode 1 presented in Figure 3b were performed for the length $L$ = 10 m.

Figure 4 shows the spatial intensity distributions of Modes 3a and 3b in the straight MOF-18 for wavelengths of 0.85 µm, 1.05 µm, 1.09 µm, and 1.13 µm. These modes have the same spatial intensity distributions in the MOF-18 core and the same polarizations, but different intensities in the core as well as in the ring gap.

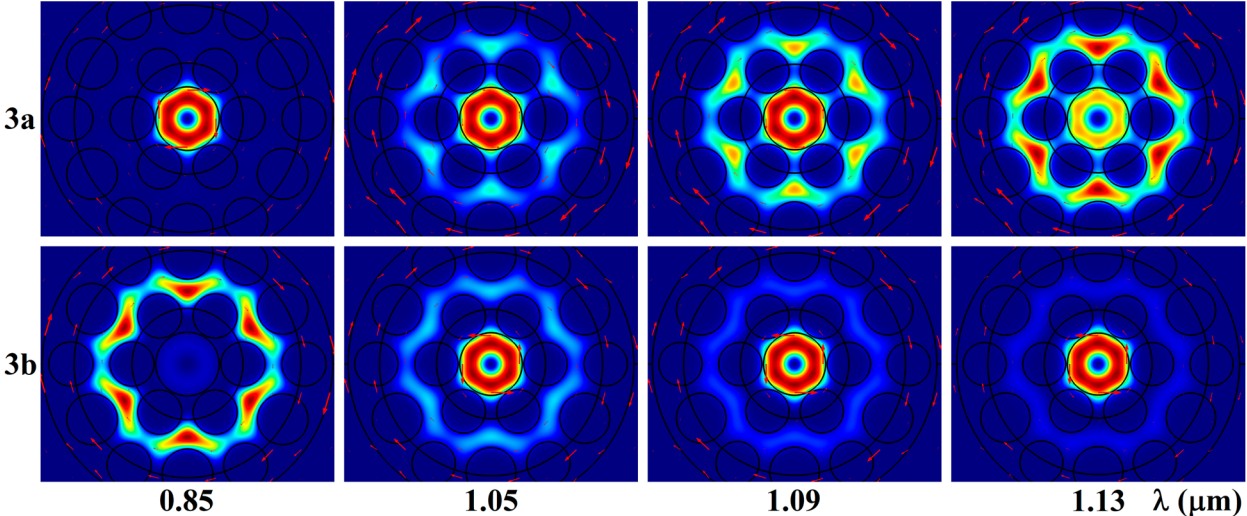

**Figure 4.** Spatial intensity distributions of Modes 3a and 3b in the straight MOF-18 for wavelengths of 0.85 µm, 1.05 µm, 1.09 µm, and 1.13 µm.

Figure 5 shows the spectral dependences of the leakage losses ($\alpha_{3a}$ and $\alpha_{3b}$) and intensities ($I_{3a}$ and $I_{3b}$) of Modes 3a and 3b in the straight MOF-18, with the intensities of modes normalized to their sum ($I_{3a} + I_{3b}$).

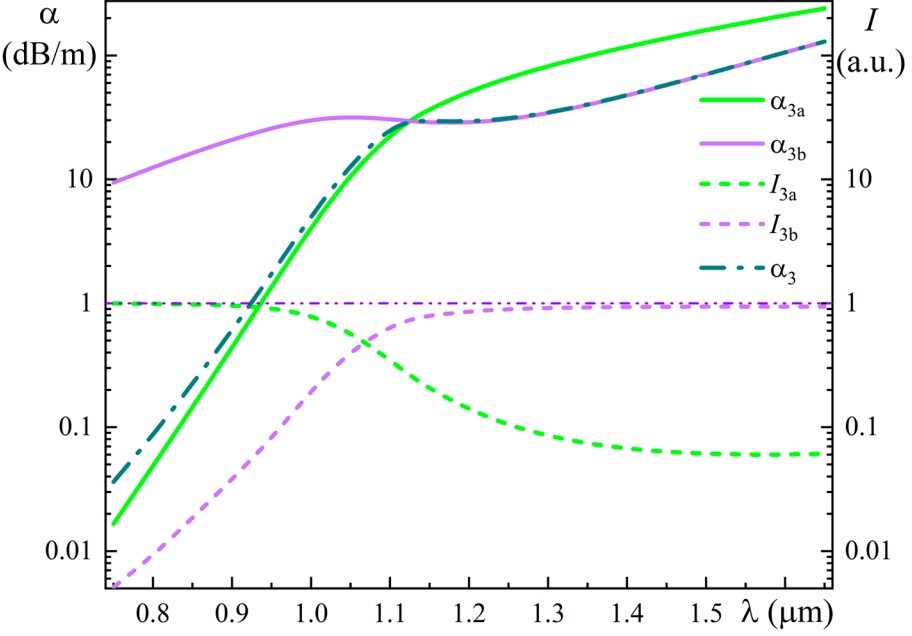

**Figure 5.** Spectral dependences of the leakage losses ($\alpha_{3a}$ and $\alpha_{3b}$) and intensities ($I_{3a}$ and $I_{3b}$) of Modes 3a and 3b in the straight MOF-18, and the resulting leakage loss for Mode 3 ($\alpha_3$).

Using the algorithm described above, we calculated the spectral dependence of the leakage loss $\alpha_3$ for Mode 3 for the length $L = 2$ m, as presented in Figure 5. Note that the length $L = 2$ m was chosen here only for clarity, since at $L = 10$ m the difference between curves $\alpha_{3a}$ and $\alpha_3$ in the short-wavelength part of the spectrum on the graph becomes indistinguishable.

Very similar results were obtained for Modes 4–6; the only noticeable differences are in the short wavelength part of the spectrum, which is associated with differences in the behavior of $I_{Mb}$ (M = 3–6) in this spectral range, while the $\alpha_{Mb}$ dependences (M = 3–6) are almost identical.

It should be noted again that for the spectral dependences of the leakage losses of fundamental modes (1, 2) and HOMs (3–6) for straight MOF-18, shown in Figure 2b, the calculations were performed for length $L = 10$ m.

### 3.2. Bent MOF-18

Calculations of leakage losses for the bent MOF-18 were carried out by replacing it with a straight MOF-18 with an equivalent refractive index profile, $n_{equ}$, which was determined (for bending along the "$x$" axis) using the expression [31]

$$n_{equ}(x,y) = n(x,y)\left(1 + \frac{x}{R}\right), \tag{7}$$

where $n(x, y)$ is the original refractive index profile of straight MOF-18, $R$ is the bending radius in meters.

Figure 6a shows the spatial intensity distributions for two polarizations of the fundamental modes (1 and 2) and four HOMs (denoted from three to six in descending order of the real part of their effective refractive index $n_{eff}$) in the bent MOF-18 at a wavelength of 0.85 μm for bending along the "$x$" axis and a radius $R_x = 0.08$ m. Figure 6b shows the spatial intensity distributions for two FMs (1, 2) and four HOMs (3–6) in the bent MOF-18 at a wavelength of 0.85 μm for bending along the "$y$" axis and a radius $R_y = 0.08$ m. In this case, all modes were designated in a different order (independent on value of the real part of their effective refractive index $n_{eff}$), namely so that for each number, the spatial intensity distributions and polarizations of the modes coincide with these parameters for the case of bending along the "$x$" axis. The reason for such a choice will be explained later. Fundamental Modes 1 and 2 are of the $LP_{01}$ type, differing in polarization; Modes 3 and 4 are of the $LP_{11o}$ type, also differing in polarization; and Modes 5 and 6 are of the $LP_{11e}$ type, also differing in polarization [32].

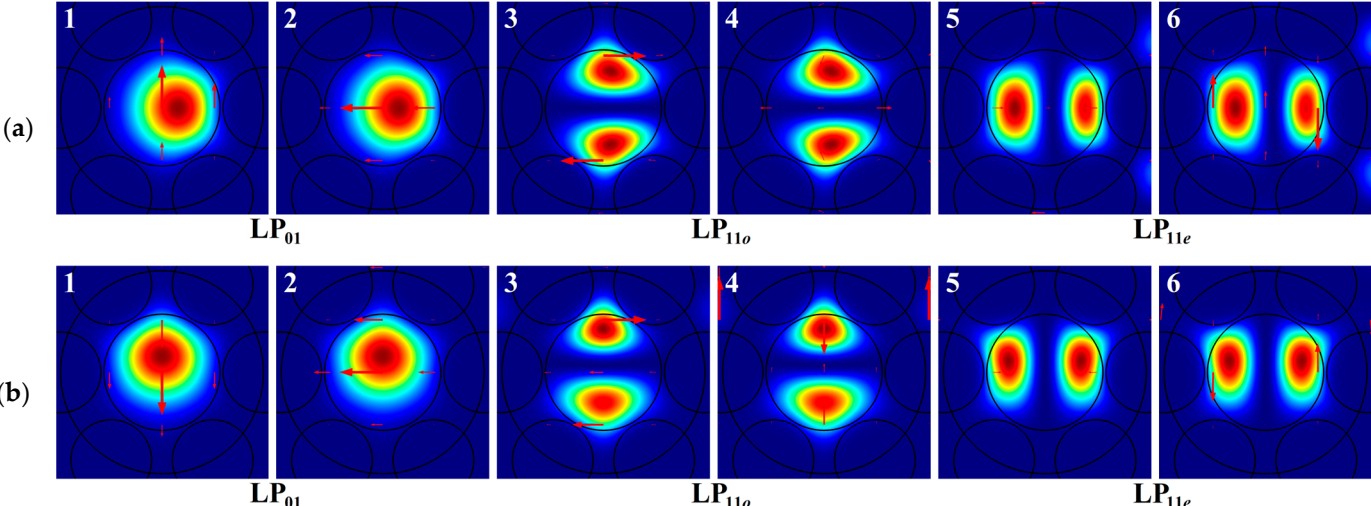

**Figure 6.** Spatial intensity distributions of (1, 2) fundamental modes and (3–6) higher-order modes in the bent MOF-18 at a wavelength of 0.85 μm for bending radii: (**a**) $R_x = 0.08$ m and (**b**) $R_y = 0.08$ m.

Figure 7 shows the spectral dependences of leakage losses of (1, 2) fundamental modes and (3–6) higher-order modes in the bent MOF-18 for bending radii: (**a**) $R_x$ = 0.08 m and (**b**) $R_y$ = 0.08 m. These spectral dependences were calculated for length $L$ = 10 m, as for the straight MOF-18. As can be seen from Figure 7, the spectral range of the single-mode regime for the bent MOF-18 for bending along the "*x*" axis is from 0.91 μm to 1.22 μm, and for bending along the "*y*" axis is from 0.93 μm to 1.21 μm.

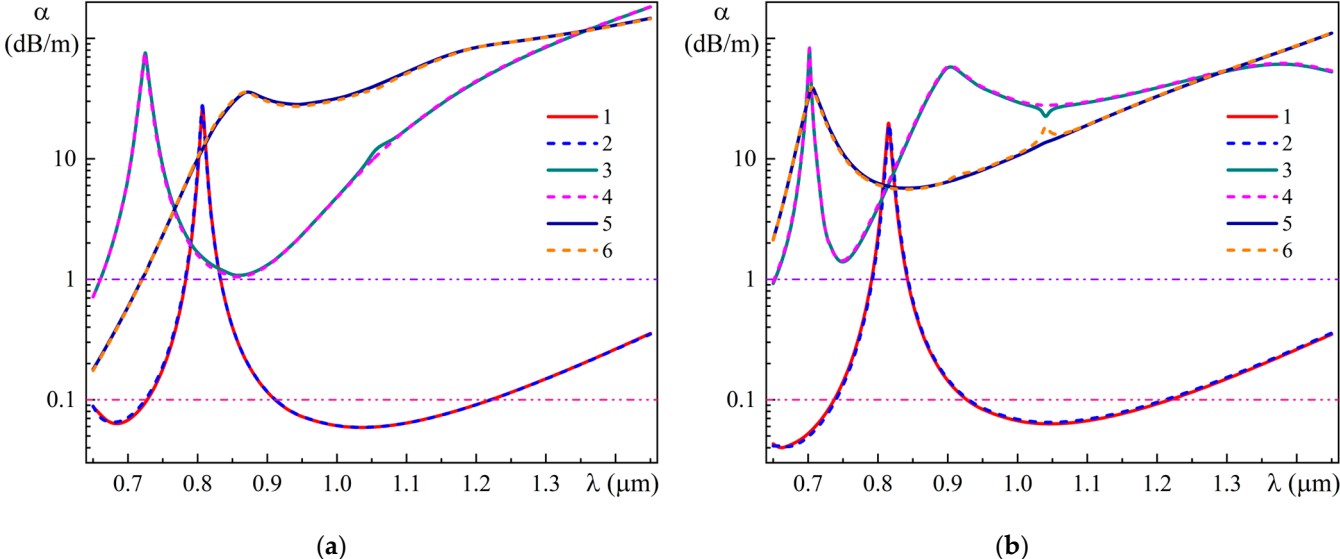

(**a**)             (**b**)

**Figure 7.** Spectral dependences of leakage losses of (1, 2) fundamental modes and (3–6) higher-order modes in the bent MOF-18 for bending radii: (**a**) $R_x$ = 0.08 m and (**b**) $R_y$ = 0.08 m.

One of the very remarkable features of the spectral dependences of the leakage losses of FMs 1 and 2 is the presence of noticeable maxima near the wavelength of 807 nm (for MOF-18 bending along the "*x*" axis) or 814 nm (for bending along the "*y*" axis). Although they are located far enough from the spectral region of interest (around 1.05 μm), it is necessary to clearly understand the mechanism of their occurrence to take into account and introduce possible adjustments to the parameters of MOF-18 in further calculations, as well as for some other tasks.

Although these maxima are similar in shape to some resonance dependencies, in reality, this loss behavior is due to the presence of two modes, 1a and 1b (as well as 2a and 2b), which have sharp changes in their losses in opposite directions in a relatively narrow spectral range near these wavelengths (807 nm or 814 nm). Figure 8 shows an example of the spatial intensity distributions of Modes 1a and 1b in the bent MOF-18 for a bending radius of $R_x$ = 0.08 m for wavelengths of 801 nm, 805 nm, 807 nm, 810 nm, and 814 nm.

Figure 9 shows the spectral dependences of the leakage losses ($\alpha_{1a}$ and $\alpha_{1b}$) and intensities ($I_{1a}$ and $I_{1b}$) of Modes 1a and 1b in the bent MOF-18 for a bending radius of $R_x$ = 0.08 m. Using the algorithm described above, we calculated the spectral dependence of the leakage loss $\alpha_1$ for Mode 1 for the length $L$ = 2 m, as presented in Figure 9. Note that the length $L$ = 2 m was chosen here only for clarity, since at $L$ = 10 m, the difference between curves $\alpha_{1a}$ and $\alpha_1$ in the long-wavelength part of the spectrum on the graph becomes indistinguishable. It is worth noting that, as can be seen from Figure 9, only in a very narrow spectral range (approximately from 0.79 μm to 0.82 μm) do the intensities of $I_{1a}$ and $I_{1b}$ simultaneously account for more than 10% of the total sum.

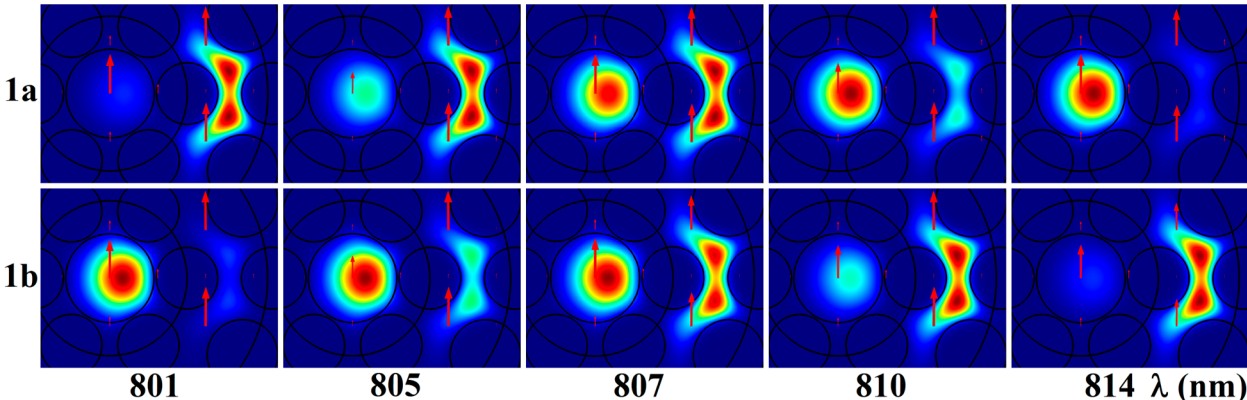

**Figure 8.** Spatial intensity distributions of Modes 1a and 1b in the bent MOF-18 for a bending radius of $R_x = 0.08$ m for wavelengths of 801 nm, 805 nm, 807 nm, 810 nm, and 814 nm.

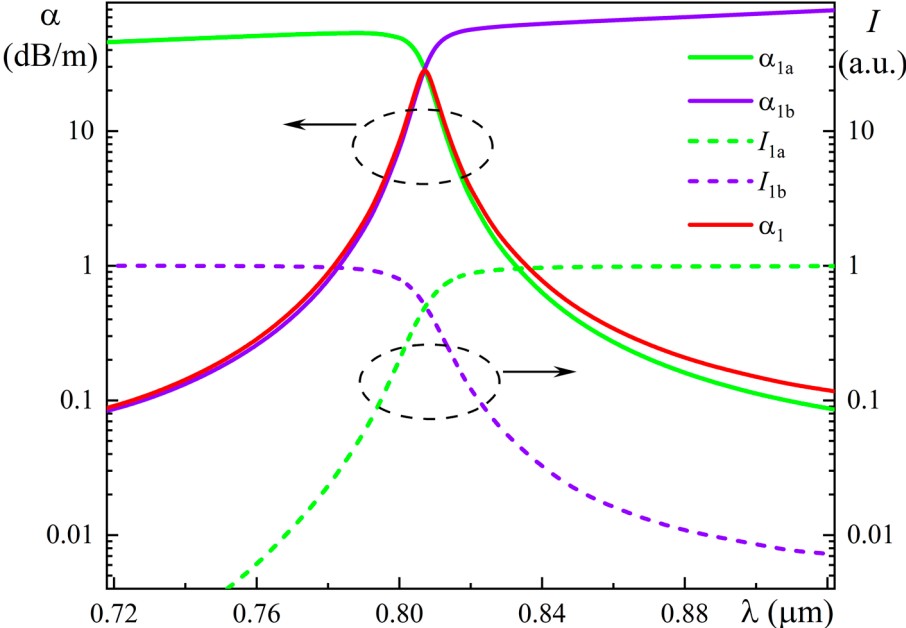

**Figure 9.** Spectral dependences of the leakage losses ($\alpha_{1a}$ and $\alpha_{1b}$) and intensities ($I_{1a}$ and $I_{1b}$) of Modes 1a and 1b in the bent MOF-18 for bending radius $R_x = 0.08$ m, and the resulting leakage loss for FM 1 ($\alpha_1$).

In practice, this means that Modes 1a and 1b simultaneously have a visually noticeable intensity in the core of MOF-18 only in this interval, if estimated by their spatial intensity distributions, which can be clearly seen in Figure 8. Outside this wavelength range, in the long-wavelength region of the spectrum, Mode 1a looks like the usual fundamental mode of a bent MOF, and Mode 1b looks like a pure cladding mode, i.e., the ring gap mode. In addition, in the short-wavelength region of the spectrum, Mode 1b looks like the usual fundamental mode of a bent MOF, and Mode 1a looks like a pure ring gap mode.

Since all these changes occur in a narrow spectral range, they can be conventionally described as a quasi-resonant transformation of the core Mode 1a into the cladding mode as the wavelength decreases, accompanied by the transformation of the cladding Mode 1b into the core mode. At the same time, in the long-wavelength region of the spectrum, the relative intensity of Mode 1b in the core does not decrease below 1% of the sum, and at wavelengths longer than 1.2 μm, it even begins to increase smoothly.

In addition, an important factor in the correct assignment of modes was careful control of the changes in the effective refractive index of all modes, especially in the region of sharp

changes in the leakage losses. Figure 10a shows the spectral dependences of the difference of refractive indices of pure silica glass and the effective refractive indices of Modes 1a and 1b, $n_{sil}-n_{1a}$ and $n_{sil}-n_{1b}$ for a bent MOF-18 with a bending radius of $R_x$ = 0.08 m. For greater clarity, Figure 10b shows the spectral dependence of the difference $n_{1a}-n_{1b}$ of the effective refractive indices of Modes 1a and 1b.

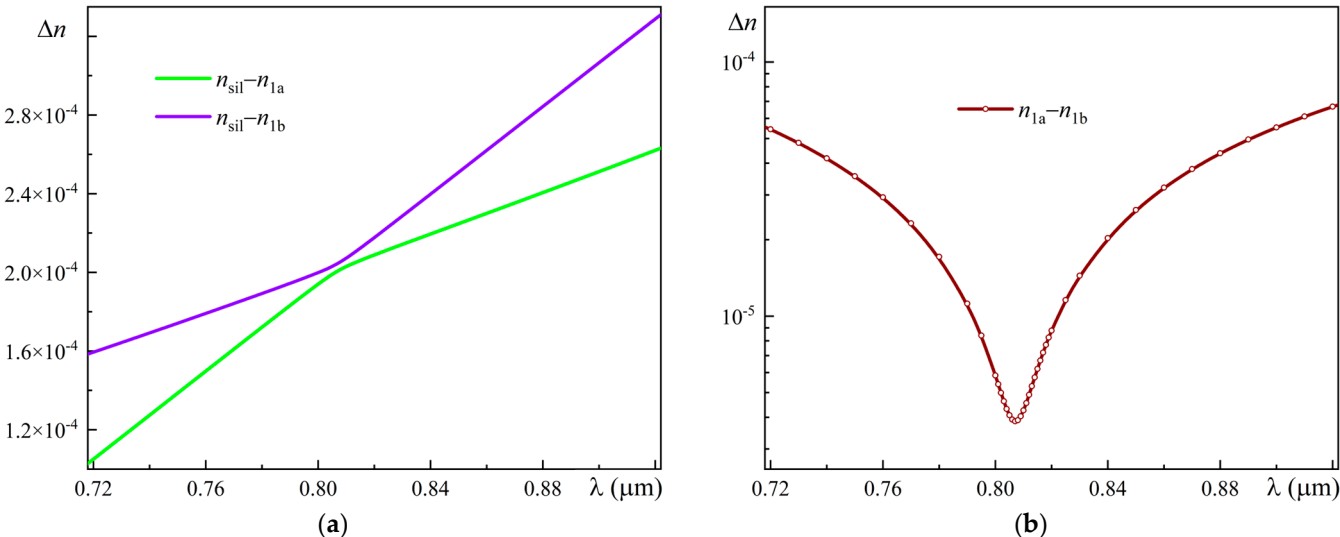

**Figure 10.** Spectral dependences for a bent MOF-18 with a bending radius of $R_x$ = 0.08 m: (**a**) of the difference between the refractive indices of pure silica glass and the effective refractive indices of Modes 1a and 1b: $n_{sil}-n_{1a}$ and $n_{sil}-n_{1b}$; (**b**) of the difference $n_{1a}-n_{1b}$ between the effective refractive indices of Modes 1a and 1b.

This behavior of the effective refractive indices $n_{1a}$ and $n_{1b}$ can be explained qualitatively using the Kramers–Kronig relations, which reflect the causality principle and, in a particular case, mean that absorption (loss) at some wavelength leads to a non-unity refractive index and vice versa. In [33], the Kramers–Kronig relations for the effective refractive indices of modes in an optical fiber were obtained. The authors of [33] showed that if the dispersion and absorption of the material in a spectral range of interest can be neglected, then the attenuation (leakage) loss appears in these relations as an effective loss term. Thus, we can suggest that changes in the leakage loss led to corresponding variations in refractive indices. Since the spectral dependences of the leakage losses for Modes 1a and 1b in the region around 807 nm have an opposite character of change (Figure 9), the real parts of the effective refractive indices for these modes ($n_{1a}$ and $n_{1b}$) also change in the opposite directions (Figure 10a). As a result, the spectral dependence for the difference $n_{1a}-n_{1b}$ has the form shown in Figure 10b.

A very similar situation is observed for Mode 2, which can be easily explained given the almost complete coincidence of the spectral dependences of the leakage losses of Modes 1 and 2, shown in Figure 7a. In the case of MOF-18 bending along the "*y*" axis, there are slight differences in the position (814 nm) and peak height for Modes 1 and 2 (which are also almost identical), as can be seen in Figure 7b.

The higher-order Modes 3–6 for the bent MOF-18 have appreciable intensities in the ring gap (as for the straight MOF-18) in almost the entire spectral range studied (0.65–1.45 μm), but have a specific character of the spatial intensity distribution in the ring gap. In addition, they have several prominent maxima, which are of the same nature as in the examples of Modes 1a and 1b considered above.

Figure 11 shows an example of the spatial intensity distributions of Modes 3a and 3b in the bent MOF-18 for a bending radius of $R_x$ = 0.08 m for wavelengths of 718 nm, 720 nm, 724 nm, 728 nm, and 730 nm. As can be seen from this figure, Modes 3a and 3b in this relatively narrow spectral range (near the wavelength of 724 nm) have sharp changes in

intensities in the core and in the ring gap. In particular, in the long-wavelength region of the spectrum, Mode 3a has a maximum intensity value in the core and a minimum in the ring gap, which change in opposite directions, so that in the short-wavelength region of the spectrum, Mode 3a has a minimum intensity value in the core and a maximum value in the ring gap. For Mode 3b, the picture has an opposite character. That is, the observed character of changes is close to that observed for Modes 1a and 1b (see Figure 8), with the difference that intensities of Modes 3a and 3b both in the core and in the ring gap have appreciable intensity in almost the entire spectral range studied.

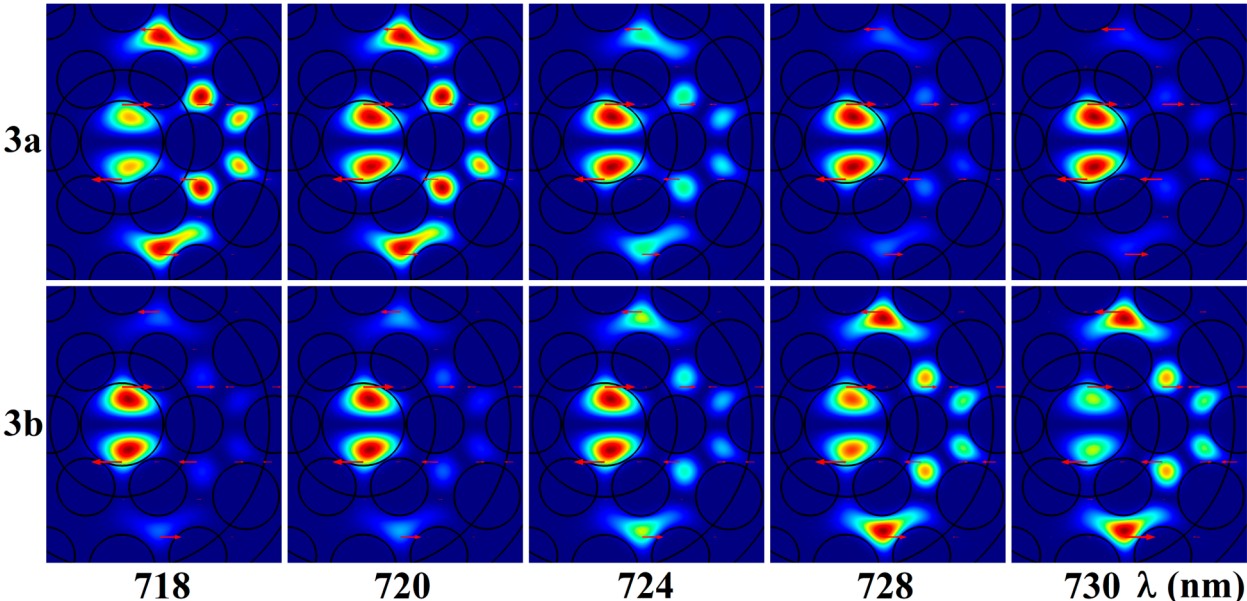

**Figure 11.** Spatial intensity distributions of Modes 3a and 3b in the bent MOF-18 for a bending radius of $R_x$ = 0.08 m for wavelengths of 718 nm, 720 nm, 724 nm, 728 nm, and 730 nm.

The spectral dependences of leakage losses ($\alpha_{3a}$ and $\alpha_{3b}$) and intensities ($I_{3a}$ and $I_{3b}$) of Modes 3a and 3b in bent MOF-18 for a bending radius of $R_x$ = 0.08 m have characters that are very similar to those of the dependences for Modes 1a and 1b (Figure 9), with minor differences. A very similar situation is observed for Mode 4, which can be easily explained given the almost complete coincidence of the spectral dependences of the leakage losses of Modes 3 and 4, shown in Figure 7a.

As for Modes 5 and 6, the spectral dependences of leakage losses for them, shown in Figure 7a, are very similar to those for straight MOF-18, shown in Figure 2b, with minor differences. Accordingly, the spectral dependences of leakage losses ($\alpha_{5a}$ and $\alpha_{5b}$) and intensities ($I_{5a}$ and $I_{5b}$) of Modes 5a and 5b in bent MOF-18 for a bending radius of $R_x$ = 0.08 m have characters that are very similar to those of the dependences for these modes in straight MOF-18, shown in Figure 5.

In the case of MOF-18 bent along the "$y$" axis, there are some differences in the spectral dependences of leakage losses of HOMs. In particular, the spectral dependences of the leakage losses for Modes 5 and 6, shown in Figure 7b, are generally similar to those for Modes 3 and 4 of MOF-18 bent along the "$x$" axis, shown in Figure 7a, with some differences in the position (705 nm), height, and width of the observed peak.

As for Modes 3 and 4, they have two pronounced maxima (702 nm and 900 nm) in the spectral dependences of the leakage losses for them, shown in Figure 7b, which is explained by the presence of three modes Ma, Mb, and Mc (M = 3, 4) in the spectral range under study.

Figure 12a shows the spatial intensity distributions of Modes 3a, 3b, and 3c in the bent MOF-18 for bending radius $R_y$ = 0.08 m for wavelengths of 702 nm, 860 nm, and 900 nm, and Figure 12b shows the spectral dependences of the leakage losses $\alpha_{3a}$, $\alpha_{3b}$, and $\alpha_{3c}$

of these modes in the bent MOF-18. Figure 12c shows the spectral dependences of the difference between the refractive indices of pure silica glass and the effective refractive indices of Modes 3a, 3b, and 3c: $n_{sil}-n_{3a}$, $n_{sil}-n_{3b}$, and $n_{sil}-n_{3c}$ for a bent MOF-18 with a bending radius of $R_y = 0.08$ m. For greater clarity, Figure 12d shows the spectral dependence of the difference $n_{3b}-n_{3c}$ between the effective refractive indices of Modes 3b and 3c. Thus, it becomes clear that the dependences for $n_{sil}-n_{3b}$ and $n_{sil}-n_{3c}$ do not intersect near the wavelength of 702 nm, although from Figure 12c it is impossible to understand this fact.

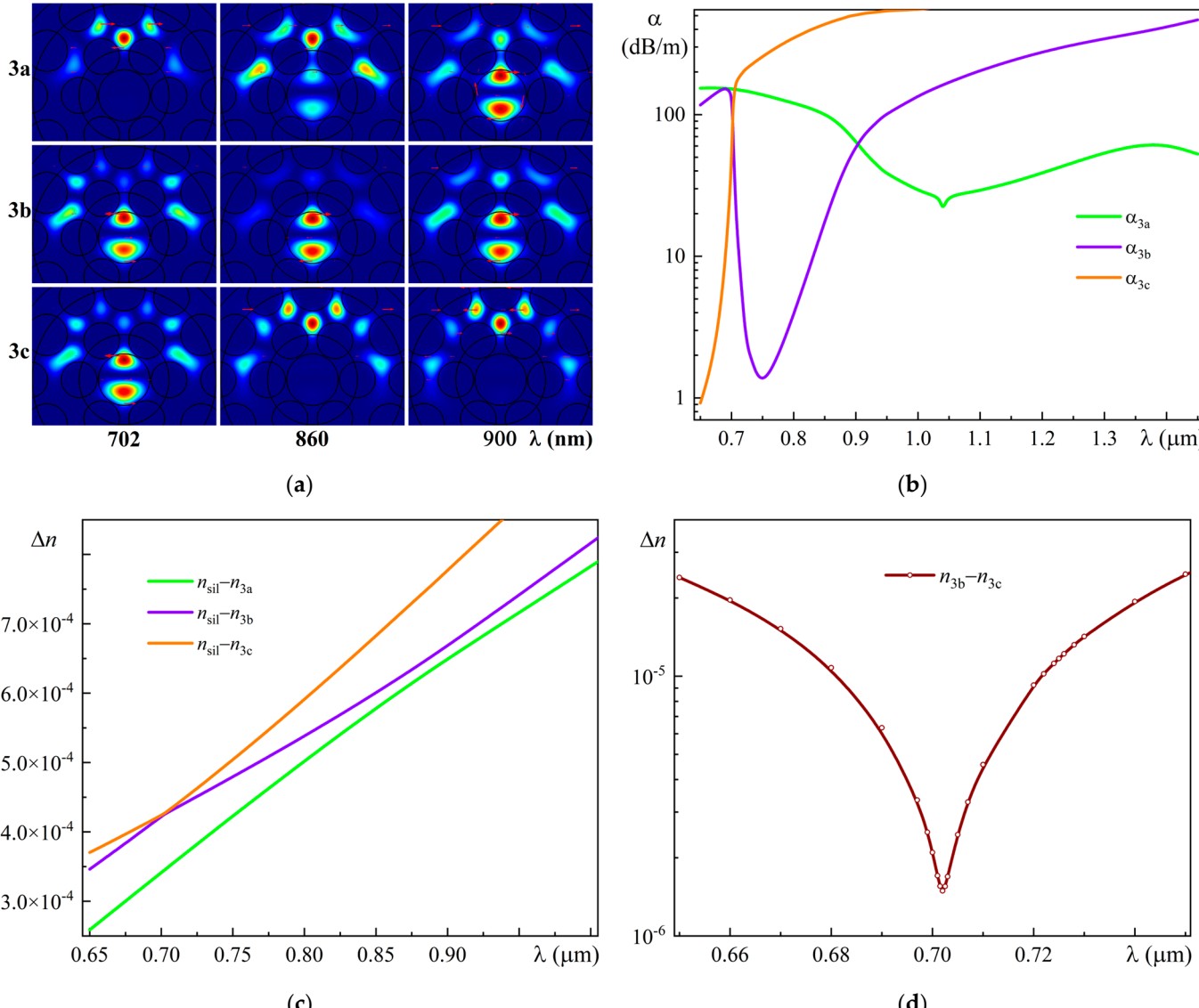

**Figure 12.** (**a**) Spatial intensity distributions of Modes 3a, 3b, and 3c in the bent MOF-18 for bending radius $R_y = 0.08$ m for wavelengths of 702 nm, 860 nm, and 900 nm; (**b**) Spectral dependences of the leakage losses $\alpha_{3a}$, $\alpha_{3b}$, and $\alpha_{3c}$ of Modes 3a, 3b, and 3c in the bent MOF-18 for $R_y = 0.08$ m; Spectral dependences for a bent MOF-18 with $R_y = 0.08$ m: (**c**) of the difference between the refractive indices of pure silica glass and the effective refractive indices of Modes 3a, 3b, and 3c: $n_{sil}-n_{3a}$, $n_{sil}-n_{3b}$, and $n_{sil}-n_{3c}$; (**d**) of the difference $n_{3b}-n_{3c}$ between the effective refractive indices of Modes 3b and 3c.

The final spectral dependences of leakage losses for Modes 3–6, shown in Figure 7, were also obtained using the previously described algorithm for length $L = 10$ m.

Since in practice it is impossible to strictly control the orientation of the internal MOF-18 structure relative to the bending direction, it is necessary to estimate the leakage losses averaged over the different bending directions and, for unpolarized radiation, also over the

polarizations. The accurate calculation of these averaged parameters is rather laborious, since it requires calculations for many bending directions with a small angle step.

As a first step in such an estimation, an averaging can performed for the two bending directions for which the calculations were carried out, i.e., for the "*x*"-axis and "*y*"-axis bends. In order to clearly define the algorithm for such averaging, let us turn to Figure 6, which shows the spatial intensity distributions for two polarizations of the fundamental mode (1 and 2) and four HOMs (from 3 to 6) in the bent MOF-18 for different bending directions. As can be seen from this figure, for each mode number M (1–6), the spatial distributions of intensity and polarization coincide quite well for bends along orthogonal directions; only a slight transformation occurs. However, the values of leakage losses at a particular wavelength for each of these modes may differ significantly for different bending directions.

Therefore, the previously considered algorithm for determining the resulting losses needs to be slightly corrected: first, we need to consider the propagation of radiation with the initial intensity $I_{\text{Min1}}$ along the MOF-18 section of length $L/2$, where losses for each mode correspond to losses for the case of bending along the "*x*" axis (or vice versa, "*y*"), and determine the intensity at the output of this section, $I_{\text{Mout1}}$. Then, it is necessary to consider the propagation of radiation with the initial intensity $I_{\text{Min2}} = I_{\text{Mout1}}$ along another MOF-18 section of length $L/2$, where losses for each mode correspond to losses for the case of bending along the "y" axis (or vice versa, "x"), and to determine the intensity at the output of this section, $I_{\text{Mout2}}$. Then, we can determine the final loss for the MOF-18 section of length $L$, averaged over the two bending directions.

For the case of non-polarized radiation for each pair of modes, 1 and 2 (LP$_{01}$), 3 and 4 (LP$_{11o}$), 5 and 6 (LP$_{11e}$), the incoming radiation with the initial intensity $I_{\text{LPin1}}$ should be divided in half between the two modes of the respective pair, then the radiation propagation of each mode along two MOF-18 sections $L/2$ bended in different directions should be examined sequentially, and the resulting intensities added, providing the value $I_{\text{LPout2}}$. Then, we can determine the final loss for the MOF-18 of length $L$, averaged over the two bending directions and over two polarizations.

Figure 13 shows the spectral dependences of leakage losses in a MOF-18 for the fundamental (LP$_{01}$) and higher-order (LP$_{11o}$ and LP$_{11e}$) modes at a bending radius of 0.08 m averaged over bend directions.

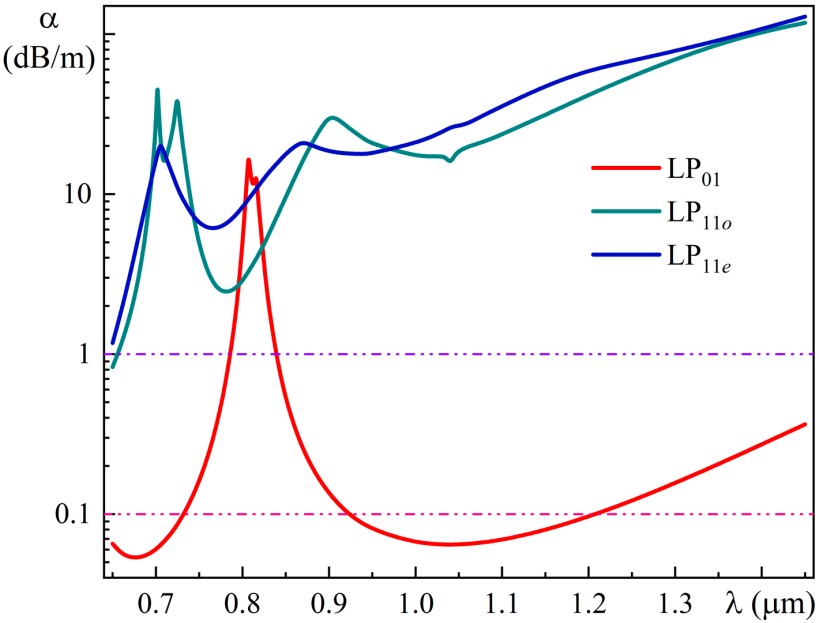

**Figure 13.** Spectral dependences of leakage losses in a MOF-18 for the fundamental (LP$_{01}$) and higher-order (LP$_{11o}$ and LP$_{11e}$) modes at a bending radius of 0.08 m averaged over bend directions.

As can be seen from Figure 13, the spectral range of the single-mode regime for the bent MOF-18 for bending radius of 0.08 m is from 0.92 μm to 1.21 μm. The leakage losses for the fundamental mode at a wavelength of 1.05 mm are 0.064 dB/m; at the same time, the losses for HOMs exceed 18.6 dB/m; that is, the ratio of leakage losses of HOMs to losses of the FM exceeds 290.

## 4. Experimental Results

The MOF-18 was fabricated in several stages. First, the drilling of a rod of pure silica glass F300 with tubular diamond drills of different diameters was carried out. Then, prepared fluorine-doped silica rods with a diameter slightly smaller than the corresponding holes were inserted into these holes, and this assembly was consolidated into a preform. At the final stage, MOF-18 was drawn from the obtained preform with the aid of the Large-scale research facilities "Fibers" (UNU Fibers) of GPI RAS.

Figure 14 shows a SEM image of the cross-section of the fabricated MOF-18 with an outer diameter of $D_{\text{fiber}}$ = 135 μm and a core diameter of $D_{\text{core}}$ = 22.5 μm, which was obtained using the scanning electron microscope JSM-5910LV (shared research facilities GPI RAS).

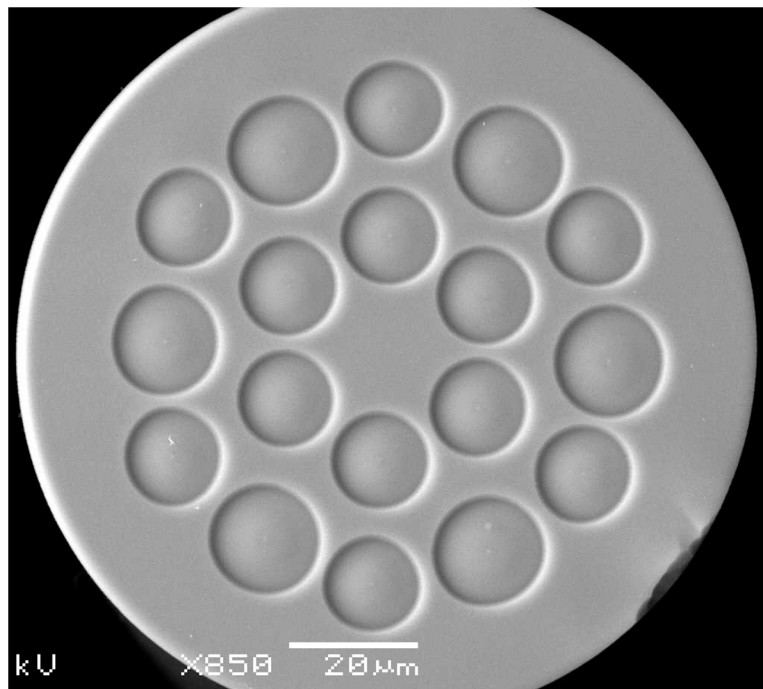

**Figure 14.** SEM image of the cross-section of the fabricated MOF-18; $D_{\text{fiber}}$ = 135 μm; $D_{\text{core}}$ = 22.5 μm.

Figure 15a shows the measured losses for this sample of fabricated MOF-18 at a bending radius of 0.1 m in a free coil. They were less than 0.1 dB/m in the wavelength range from 0.9 to 1.5 μm, while in the wavelength range from 1.0 to 1.1 μm, these losses were from 0.03 dB/m to 0.02 dB/m.

The experimental test showed that the segments of this MOF-18 longer than 5 m are single-mode. Inset in Figure 15a shows photos of the observed intensity in the near-field of MOF-18 at different displacements of the incoming beam (for radiation with a wavelength of 0.976 μm).

Figure 15b shows the dependences of the measured losses of the fabricated MOF-18 on the bending value when varying the distance *L* between the two walls, limiting the free coil of MOF-18. Qualitatively, these dependences are quite consistent with the results of our calculations performed earlier [26]: first, we observe a shift of the loss maximum to the long-wavelength region of the spectrum with decreasing *L* (Figure 15b), which is actually

equivalent to decreasing the bending radius of MOF-18 ([26], Figure 13); second, we also observe an increase in the loss value at the maximum of the observed band with decreasing $L$, that is, with decreasing bending radius.

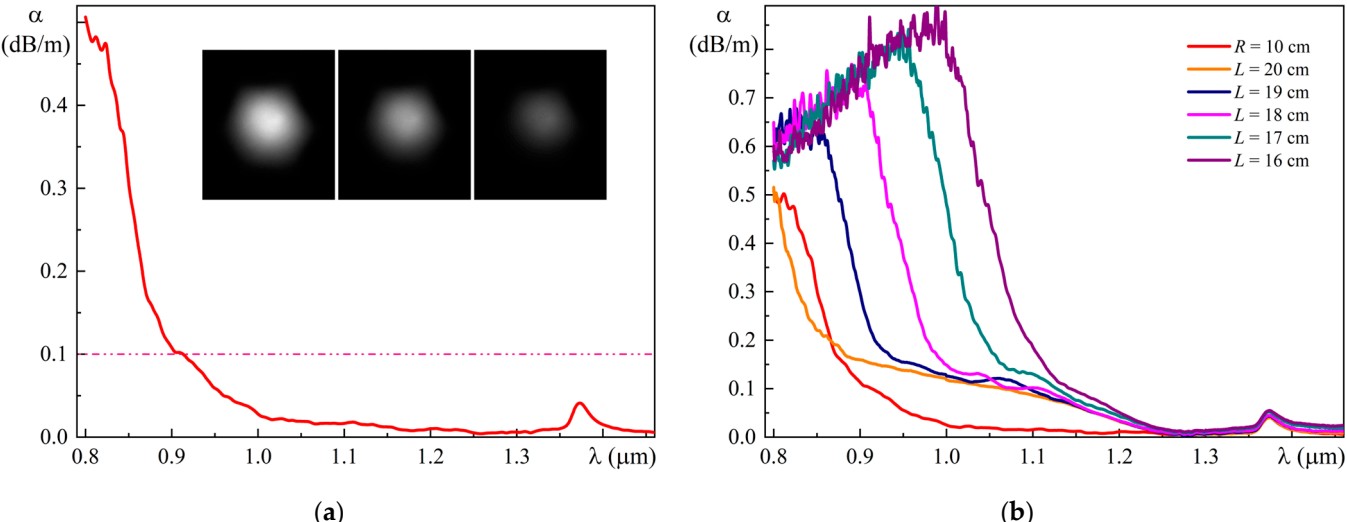

**Figure 15.** (**a**) Measured losses of the fabricated MOF-18 at a bending radius of $R = 0.1$ m. Inset–observed intensity in the near-field of MOF-18 at different displacements of the incoming beam; (**b**) Dependences of losses of the fabricated MOF-18 from bending when varying the distance $L$ between the two walls, limiting the free coil with MOF-18.

As for the differences in the loss value and spectral band width, they are due to some deviations in the parameters of the fabricated MOF-18 structure from the parameters of the model structure chosen for the calculations (see Figure 14). This led, in particular, to a small increase in the bending radius of the MOF-18 (0.1 m instead of 0.08 m for the model MOF-18 structure), for which the single-mode regime and small bending losses for the fundamental mode are provided.

The main factor that affects the permissible value of the bending radius of MOF-18 is the presence of the leakage loss maximum (near the wavelength of 0.81 mm, see Figure 13) of the fundamental mode. The position of this maximum (at a fixed bending radius) depends largely on the width of the gap between the layers of fluorine-doped silica glass elements. The gap between the elements of diameter $d_2$ of the second layer and the elements of the first layer is $Z_{12} = \Lambda_2 - d_1/2 - d_2/2$, and the gap between the elements of diameter $d_3$ of the second layer and the elements of the first layer is $Z_{13} = \Lambda_3 - d_1/2 - d_3/2$. The size of the gap between the elements of the first layer $Z_{11} = \Lambda_1 - d_1$. In our calculations, we used values of parameters $Z_{12}/Z_{11} = 1.81$ and $Z_{13}/Z_{11} = 1.50$, whereas the measured parameters of the fabricated MOF-18 (from Figure 14) turned out to be larger: $Z_{12}/Z_{11} = 2.10$ and $Z_{13}/Z_{11} = 1.71$.

At the same time, in contrast to the calculated data (Figure 13), the experiment does not show an increase in losses in the long-wavelength part of the spectrum. The latter fact can be explained by a slightly larger value of the parameter $d_1/\Lambda_1 = 0.840$ (from Figure 14), while the calculations were performed for a smaller value of $d_1/\Lambda_1 = 0.795$. Accordingly, at a larger value of $d_1/\Lambda_1$, the value of the gaps $Z_{11}$ between the elements of the first layer is smaller, which leads to a reduction in the leakage losses of the fundamental mode.

## 5. Discussion

There are some theoretical studies of different types of MOFs with a large mode area and single-mode guidance that consider the suppression of higher-order modes in terms of the coupling of these modes of the core with the leaky modes of the cladding [34–36]. The authors of these works believe that this coupling is due to the coincidence of the real parts

of their effective refractive indices (index-matched coupling), that is, the intersection of the spectral dependence of the effective refractive index of the higher-order mode of the core with the dependence of the effective refractive index of the leaky mode of the cladding at some wavelength. At the same time, no physical mechanisms were provided that could explain this coupling between these modes. Although this approach can help determine some parameters of MOF structures that increase leakage losses of higher-order modes, the question of physical mechanisms that could bring clarity to such an effect remains open.

The results presented in this paper show that both the fundamental mode and the higher-order modes of the straight and bent MOF-18 have additional local maxima of intensity in the ring gap. In addition, the fundamental mode and the higher-order modes (denoted by the number M from 1 to 6) are in some sense (conditionally) degenerate since they have a group of two or three close modes, and for each group, these modes have very similar spatial intensity distributions in the core of MOF-18. In addition, for each group, these modes have different ratios of intensities in the core and in the ring gap (varying with wavelength), different levels of leakage losses, and different real parts of their effective refractive index $n_{eff}$ (we have designated them as Ma, Mb, and Mc in descending order of their $n_{eff}$).

It is worth noting that, in the classical sense, the fundamental mode in an ordinary optical fiber is twice degenerate because it has two different (orthogonal) polarizations, the same spatial intensity distributions in the core, and the same values of the real parts of the effective refractive indices [28] (p. 30).

One of the remarkable features of the spectral dependence of the leakage losses of the fundamental mode and HOMs of MOF-18 is the presence of noticeable maxima near certain wavelengths. These maxima may be conventionally described in terms of the quasi-resonant transformation of the core modes Ma (M = 1–6) into the cladding modes as the wavelength decreases, accompanied by the transformation of the cladding modes Mb into the core mode. Such transformations also include sharp changes in their losses in opposite directions in a relatively narrow spectral range near these certain wavelengths. As a result, the final losses that were calculated according to the algorithm described above have pronounced maxima at these certain wavelengths.

A similar situation was described by us in [37], but there we considered a MOF-30 with air holes in the cladding and quasi-resonance transformation of higher-order core modes Ma (M = 3–6) into cladding modes as the wavelength increases, accompanied by the transformation of cladding modes Mb into higher-order core modes.

As for the experimental data, we have already noted some deviations in the parameters of the fabricated MOF-18 structure from the parameters of the model structure. In particular, these deviations lead to differences in the loss value and spectral band width between the measured losses of the fabricated MOF-18 (Figure 15b) and the calculated leakage losses of the fundamental mode of the bent MOF-18 (Figure 13). These differences may be decreased (or even eliminated) by taking into account real parameters of the fabricated MOF-18 structure (from Figure 14) and performing calculations of the leakage losses with detailed averaging over the different bending directions and also over the polarizations. The previously considered algorithm for averaging over the two bending directions may be easily modified for averaging with sufficiently detailed step of the bending direction. This is one of the tasks for our future calculations.

It is worth noting that there are several ways to achieve better MOF parameters for any particular problem. First, we may use just the same MOF-18 structure but select another ratio of element diameters $d_2/d_1$ and $d_3/d_1$, since our current choice (1.15 and 1.00, respectively) was due to the particular tubular diamond drills for drilling silica glass that we have, but that is not a problem. Then, to determine the best MOF-18 parameter values, we should use a multi-objective optimization algorithm (e.g., see [27]) or machine learning technology. Second, we may add some additional rings of elements with different diameters. The simplest variant has 12 additional elements with a diameter of $d_4$ that are

located opposite the bridges between the elements of the second ring and at a distance of $\Lambda_5$ from these elements (somewhat similar to MOF-30 from [37]).

## 6. Conclusions

A detailed theoretical study of the properties of the original MOF-18 design presented in this paper was performed considering all modes with relatively low leakage losses, taking into account their polarization state, and with detailed wavelength steps down to 1 nm (see, for example, Figures 8, 10b and 12d). The leakage losses for the fundamental and higher-order modes were calculated in the spectral range from 0.75 μm to 1.65 μm for straight MOF-18 and in the spectral range from 0.65 μm to 1.45 μm for bent MOF-18. Simulation results show that the proposed MOF-18 has single-mode guidance in the spectral range of 0.92 μm to 1.21 μm with a bending radius of down to 0.08 m.

The proposed MOF-18 was successfully fabricated and experimentally investigated. The measured losses of the fabricated MOF-18 with a core diameter of 22.5 μm and a bending radius of 0.1 m were less than 0.1 dB/m in the spectral range from 0.9 μm to 1.5 μm, while in the wavelength range from 1.0 to 1.1 μm, these losses were from 0.03 dB/m to 0.02 dB/m. It was demonstrated that the segments of this MOF longer than 5 m are single-mode. MOF-18 can be used for delivering high-power laser radiation and, in the presence of an active core, in high-power fiber lasers and amplifiers.

**Author Contributions:** Conceptualization, A.D., S.S.; methodology, M.L., V.V., A.K.; software, A.D., V.D.; validation, A.D., S.S., A.K., V.D., M.L., V.V., S.Z.; formal analysis, A.D., V.D., A.K., M.L., V.V., S.Z., S.S.; investigation, A.K., S.Z., V.V., A.D.; resources, S.S., M.L.; data curation, A.D.; writing—original draft preparation, A.D.; writing—review and editing, A.D., S.S., M.L., V.V., A.K., S.Z., V.D.; visualization, A.D., V.D.; supervision, S.S.; project administration, S.S.; funding acquisition, S.S. All authors have read and agreed to the published version of the manuscript.

**Funding:** This research was funded by the Ministry of Science and Higher Education of the Russian Federation (Grant No. 075-15-2022-315 for creation and development of the World-Class Research Center "Photonics Center").

**Institutional Review Board Statement:** Not applicable.

**Informed Consent Statement:** Not applicable.

**Data Availability Statement:** Not applicable.

**Acknowledgments:** The authors are grateful to the staff of the Large-scale research facilities "Fibers" (UNU Fibers) of GPI RAS for the fabrication and characterization of the used fibers. The authors are also grateful to the Shared research facilities GPI RAS, in particular Lyudmila D. Iskhakova, for preparing SEM images of the cross section of the fabricated MOF-18 using the Scanning Electron Microscope JSM-5910LV.

**Conflicts of Interest:** The authors declare no conflict of interest.

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
