# Peer review of "All-Glass Single-Mode Leakage Channel Microstructured Optical Fibers with Large Mode Area and Low Bending Loss"

_photonics, doi:10.3390/photonics10040465_

Round 1

Reviewer 1 Report

Review

of the article All-Glass Single-Mode Leakage Channel Microstructured Optical Fibers with Large Mode Area and Low Bending Loss,

presented by authors Alexander Denisov etc.

The article presents the results of theoretical and experimental studies of MVS-18 all-glass microstructured optical fibers containing two layers of round elements made of fluorine-doped quartz glass with a reduced refractive index, different diameters and different distances between them.

In general, the article is built very logically, consistently and pleasantly read.

As some of the issues that remained after reading the article, the following can be distinguished.

1. Why do the authors propose exactly 18 elements?

2. Will fiber quality (losses, single mode maintenance) improve, if 3 or more layers of round rods are implemented instead of 2?

The article satisfies all the requirements and topics of the photonics journal and can be accepted in its present form for publication.

Reviewer

Reviewer 2 Report

This manuscript presents an FEM simulation of a passive (i.e., no active dopant) micro-structured optical fiber (MOF) the confinement loss of FM mode and HM modes and it's ratio to demonstrate that HM can be suppressed due to high confinement loss while ensuring single-mode operation thanks to very low confinement loss at bending diameter of 0.08m.  Single mode operation was achieved  for wavelength range of 0,92um-1.21um, which could be used for delivery of high power laser radiation in the form of a passive fiber. 

The format and English language was also of very high quality. The manuscript was well presented with adequate citations as well as data. I do recommend accepting this manuscript as it is. 

Author Response

Thank you very much for your comment.

Reviewer 3 Report

The author presented the detailed theoretical analysis and experimental verification of a passive microstructured optical fiber. I believe those statements are convincible and can be accepted to be published.

Author Response

Thank you very much for your comment.

Reviewer 4 Report

Authors present a leakage channel fiber for high power fiber lasers at the 1micron band. They describe the fiber design, simulation results with experimental fabrication and measurements of the fiber.

Overall, "this leakage channel fiber" concept has been introduced before by L. Dong at IMRA more than a decade ago. Since then, Dong group and their collaborators have expanded their research on this subject at Clemson University.  As a novel addition to this concept, the authors here describe a new way designing and fabricating the leakage channel fibers that yields lower losses.

I recommend publishing the manuscript. Here are some style comments for the authors:

- First paragraph of section 2 should be moved to the end of the Introduction.

Author Response

Thank you very much for your comment. We moved the first paragraph of section 2 to the end of the Introduction. Also, we removed some words from the second sentence of this paragraph (“based on the preform fabrication”) to clarify its meaning.